# Controlling the shape of 3D microstructures by temperature and light

Marc Hippler [1,2], Eva Blasco [3], Jingyuan Qu[2,4], Motomu Tanaka [5,6], Christopher Barner-Kowollik[3,7], Martin Wegener[2,4] & Martin Bastmeyer[1,8]

Stimuli-responsive microstructures are critical to create adaptable systems in soft robotics and biosciences. For such applications, the materials must be compatible with aqueous environments and enable the manufacturing of three-dimensional structures. Poly(*N*-isopropylacrylamide) (pNIPAM) is a well-established polymer, exhibiting a substantial response to changes in temperature close to its lower critical solution temperature. To create complex actuation patterns, materials that react differently with respect to a stimulus are required. Here, we introduce functional three-dimensional hetero-microstructures based on pNIPAM. By variation of the local exposure dose in three-dimensional laser lithography, we demonstrate that the material parameters can be altered on demand in a single resist formulation. We explore this concept for sophisticated three-dimensional architectures with large-amplitude and complex responses. The experimental results are consistent with numerical calculations, able to predict the actuation response. Furthermore, a spatially controlled response is achieved by inducing a local temperature increase by two-photon absorption of focused light.

[1] Zoologisches Institut, Zell- und Neurobiologie, Karlsruhe Institute of Technology (KIT), Fritz-Haber-Weg 4, 76131 Karlsruhe, Germany. [2] Institut für Angewandte Physik, Karlsruhe Institute of Technology (KIT), Wolfgang-Gaede-Straße 1, 76131 Karlsruhe, Germany. [3] Macromolecular Architectures, Institut für Technische Chemie und Polymerchemie, Karlsruhe Institute of Technology (KIT), Engesserstrasse 18, 76128 Karlsruhe, Germany. [4] Institut für Nanotechnologie, Karlsruhe Institute of Technology (KIT), Hermann-von-Helmholtz-Platz 1, 76344 Eggenstein-Leopoldshafen, Germany. [5] Physical Chemistry of Biosystems, Institute of Physical Chemistry, Heidelberg University, Im Neuenheimer Feld 253, 69120 Heidelberg, Germany. [6] Institute for Integrated Cell-Materials Science (WPI iCeMS), Kyoto University, Kyoto 606–8501, Japan. [7] School of Chemistry, Physics and Mechanical Engineering, Queensland University of Technology (QUT), 2 George Street, Brisbane, QLD 4000, Australia. [8] Institut für Funktionelle Grenzflächen, Karlsruhe Institute of Technology (KIT), Hermann-von-Helmholtz-Platz 1, 76344 Eggenstein-Leopoldshafen, Germany. Correspondence and requests for materials should be addressed to M.W. (email: martin.wegener@kit.edu) or to M.B. (email: bastmeyer@kit.edu)

Stimuli-responsive materials are key for active tunable systems[1–4]. In recent years, a large variety of material systems suitable for macroscopic[5–7] and microscopic[8–10] architectures have been investigated and extensively reviewed[11–15]. Light as a local stimulus is of particular interest because light can readily be focused to small spots, allowing for controlled local responses. For applications in soft robotics, microfluidics and biosciences[16–19], at least two conditions need to be fulfilled. First, the materials must be compatible with aqueous conditions. Second, it must be possible to generate three-dimensional micro- or nanostructures. In this context, poly(N-isopropylacrylamide) (pNIPAM) is a well-established polymer system[20–24], which has previously been used in the field of 3D printing[7,25–27]. When the temperature is only slightly elevated above its lower critical solution temperature (LCST), pNIPAM becomes hydrophobic and expels water from its inside. As a result, the material shrinks and stiffens considerably. Upon decreasing the temperature, the reverse process takes place. Homo-structures therefore show substantial and repeatable temperature-induced volume changes[25,28,29]. However, more complex actuation requires dissimilar materials that react differently with respect to the stimulus, i.e., hetero-microstructures.

In this paper, we introduce a single photoresist based on N-isopropylacrylamide, the crosslinker N,N′-methylenebisacrylamide, and a water-soluble photoinitiator. The advantages of this photoresist are twofold. First, the local properties in a three-dimensional (3D) microstructure can be tailored by the local exposure dose during 3D laser lithography (gray-tone lithography), opening the door to 3D hetero-microstructures from a single photoresist. The resulting property differences are extremely large. Our experimental results are consistent with numerical calculations which indicate a ten-fold change in the thermal expansion coefficient and the Young's modulus versus temperature based on gray-tone lithography. As examples, we demonstrate a variety of complex 3D architectures exhibiting large-amplitude and complex actuation responses. The second advantage of the present work is that a local temperature increase and hence the actuation can be induced by two-photon absorption of focused light. This aspect potentially allows for initiating local responses in three dimensions, i.e., not just on surfaces but also inside of 3D structures.

## Results

### 3D laser lithography of pNIPAM microstructures.
We initially developed a photoresist which can be used for 3D laser lithography. To be able to fabricate stable 3D structures, it is essential to add a crosslinker to the resin, here N,N′-methylenebisacrylamide (Mbis). The ratio between the monomer and the crosslinker is critical. For very small crosslinker concentrations, one can hardly write 3D structures at all. The resulting material responds to the stimulus though. In the opposite limit of large crosslinker concentrations, one can easily write arbitrary 3D structures. However, these structures hardly respond to the stimulus temperature. The optimum crosslinker concentration lies between these limits. Therefore, one initial challenge was to identify a suitable composition which is a trade-off between fabrication properties and functionality. Eventually, we found the molar ratio of NIPAM 14:1 Mbis to be a good compromise. Furthermore, the responsive photoresist contains a highly efficient photoinitiator, i.e., lithium phenyl-2,4,6-trimethylbenzoylphosphinate (LAP)[30], and acryloxyethyl thiocarbamoyl Rhodamine B as a fluorescent dye to be able to record 3D fluorescence image stacks. All components were dissolved in ethylene glycol for fabrication. Thereafter, they were transferred to water for development and kept in solution to avoid drying of the formed hydrogel.

To investigate the stimuli-responsive properties of the material, we performed a temperature-dependent mechanical analysis of pNIPAM-based hydrogel blocks produced by 3D laser lithography via atomic force microscopy (AFM). Figure 1a shows an optical micrograph of such a block with the cantilever approached to the surface. To compare the behavior of our material system to commonly used macroscopic pNIPAM-based hydrogels, we evaluated the Young's modulus of the fabricated block as a function of temperature (Fig. 1b). By increasing the temperature from $T = 22\,°C$ to $T = 43\,°C$, the lower critical solution temperature (LCST) of the material is exceeded and the material shrinks and stiffens. As a consequence, the measured Young's modulus increased by an order of magnitude. Furthermore, the results indicated that the material transition from hydrophilic to hydrophobic is not as sharp as in the case of the non-crosslinked material[31]. There, the transition occurs almost exclusively in the narrow regime between 32 °C and 33 °C. In our case, the transition is distributed over the temperature range from 28 °C and 43 °C. This finding is in agreement with previous results[28,32,33]. As a consequence, the images in this report are recorded at 20 °C and 45 °C to capture the experimental situation below and above the LCST, respectively.

The extend of swelling in this transition is visualized in Fig. 1c. The height profile was recorded by performing a line-scan with the cantilever tip from the glass surface to the pNIPAM block (line indicated in Fig. 1a). The results show that the hydrogel shrunk by a factor of 3 upon increasing the temperature and

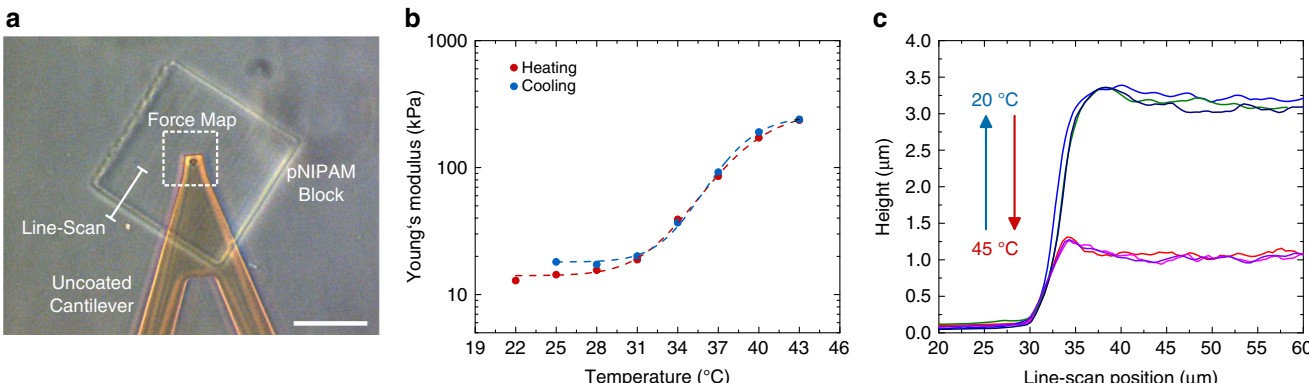

**Fig. 1** Mechanical analysis of a pNIPAM block fabricated by 3D laser lithography. **a** Optical micrograph in the AFM with overlayed indications for the force measurements and the line-scan. Scale bar is 50 μm. **b** Measured Young's Modulus as a function of temperature for a stepwise heating and cooling of the sample. **c** Height measurement via line-scanning from the glass substrate on top of the pNIPAM block. The different colors depict several cycles of heating and cooling

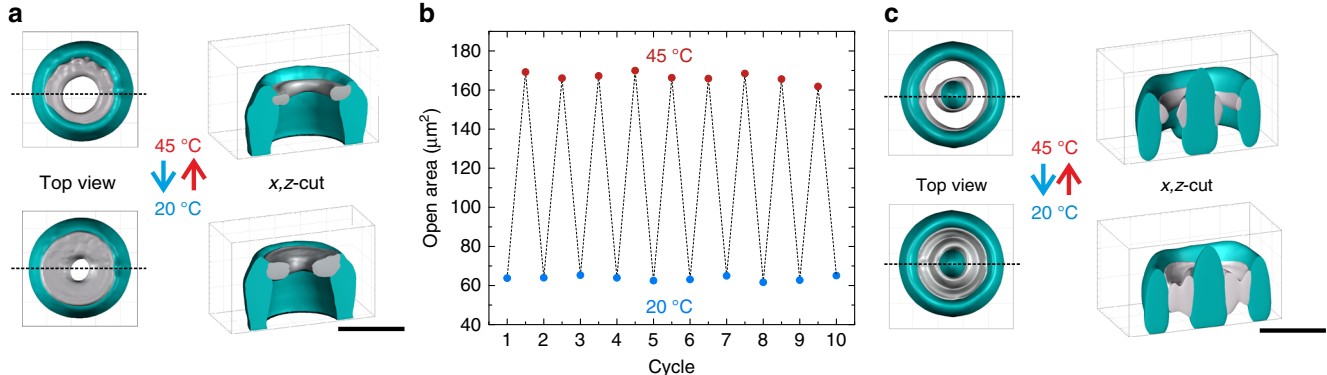

**Fig. 2** Stimuli-responsive pNIPAM valves in PETA microchannels. **a** 3D reconstruction of experimental data recorded via confocal laser scanning microscopy. Two different color channels have been recorded, allowing to separate the fluorescence from the PETA with the green fluorescent DETC and that from pNIPAM with the red fluorescent rhodamine dye. The corresponding iso-intensity surfaces are colored in turquoise and gray, respectively. Upon heating the sample to 45 °C, the opening in the middle widens. This process is reversible when cooling the sample back down. **b** Open area in the middle of the microchannel at 20 °C and 45 °C for multiple cycles of stimulation. We find no significant deterioration. **c** Alternative design with an additional inner tube and two pNIPAM-tori. Complete closure of the microchannel can be achieved in a reversible manner. Scale bars are 30 μm

re-swelled to the original height after cooling the sample back down. The graph depicts three consecutive cycles of heating and cooling for the same hydrogel block. Importantly, the response of the mechanical properties and the swelling of the microstructure was completely reversible within measurement errors.

**Combination of pNIPAM with established photoresist systems.**
3D laser lithography is a versatile technique that readily allows the combination of different photoresist systems. To highlight the potential of our resist system in this regard, we next demonstrated the combination of responsive photoresists with conventional non-responsive materials. As an example, we designed micrometer-scale rigid tubes with stimuli-responsive valves. Figure 2a shows a top view and a *x,z*-cut of a 3D reconstruction of experimental data, obtained by confocal laser scanning microscopy (LSM). In a first writing step, a common photoresist based on pentaerythritol triacrylate (PETA, colored in turquoise) was used to fabricate the tubes. Subsequently, a pNIPAM-based torus (colored in gray) was written inside the tube at a height of 20 μm. To record these images, we designed a chamber with a Peltier element to heat or cool the sample to the desired temperature.

The graph in Fig. 2b depicts the open area of the microchannel for multiple cycles of stimulation. By increasing the temperature to 45 °C, the hydrogel shrunk significantly and the opening in the middle widened. As a consequence, the measured open channel area increased by more than a factor of 2.5. Upon cooling the sample back down to room temperature the initial situation was restored. Importantly, only the pNIPAM valves reacted to the stimulus, while the structures made from PETA remain unchanged. We performed multiple cycles of heating and cooling and found no deterioration within measurement errors. An alternative design with an additional inner pNIPAM-tori is depicted in Fig. 2c. At 20 °C both tori are swollen and close the microchannel completely. At elevated temperatures the hydrogel shrinks and the microchannel opens.

**Hetero-microstructures fabricated by gray-tone lithography.**
To achieve a yet more complex response and large-amplitude actuation, more sophisticated structures with a high spatial control of the material parameters are required. In this regard, the previously shown sequential approach is inherently limited. Thus, the next step was the fabrication of responsive 3D hetero-

microstructures from a single resist and subsequent investigation of the thermo-response. Using gray-tone lithography, we created microstructures consisting of a central pillar on the glass substrate and a bi-material beam connected to it. Figure 3a shows a rendered model of such a hetero-microstructure next to a 3D reconstruction of experimental data, obtained by LSM. As discussed previously, by increasing the temperature from $T = 20$ °C to $T = 45$ °C, the LCST of the material is exceeded and the material shrinks and stiffens. The magnitude of this effect, however, is largely dependent on the crosslinking density of the polymerized hydrogel. If more fixed crosslinks between the polymer chains are present, the material is strongly confined to the geometry of the fabrication design. As a result, the material showed a much smaller thermo-response compared to a weakly crosslinked hydrogel. By increasing the temperature, the beam indicated in green (lower crosslinking density) showed a considerably stronger shrinking than the gray beam (higher crosslinking density), which led to a pronounced bending to the left-hand side.

In order to realize hetero-structures in a single fabrication step, it is necessary to achieve control over the material properties. Therefore, we investigated the effect of the exposure dose during fabrication on a bi-material beam. We exploited the flexibility of 3D laser lithography to vary the local exposure dose during writing. This gray-tone lithography approach allows us to realize materials with substantially different properties in one fabrication step from a single photoresist formulation. As usual, the notion gray-tone lithography refers to gradually changing the material properties by continuously varying the exposure dose during the process[34]. In this way, we achieved a highly localized control over the crosslinking density and consequently of the thermo-response of the structure. The relation between the exposure dose and the beam actuation was carefully assessed and hetero-microstructures were prepared by varying the fabrication parameters (Supplementary Fig. 1).

In the next step, we compared the behavior of our hetero-microstructures to the previously investigated homo-structures. Thus, we evaluated the curvature of the beams as a function of temperature (Fig. 3b, Supplementary Movie 1). The radius was obtained by fitting a circle to the optical micrographs for the respective temperatures. Clearly, the same trend as depicted in Fig. 1b becomes apparent with the strongest response in the regime between 28 °C and 40 °C.

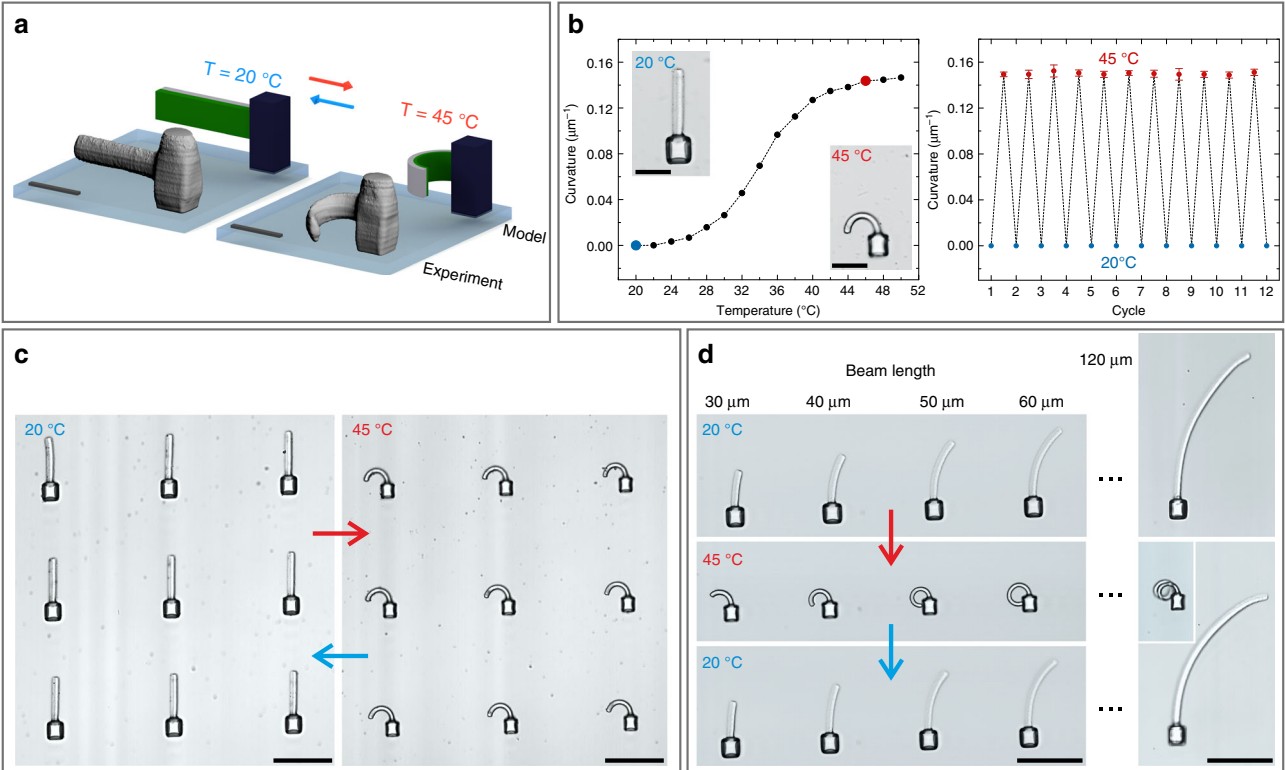

**Fig. 3** Temperature-induced actuation using pNIPAM-based hetero-microstructures. **a** Scheme of bi-material hetero-structures with the two materials highlighted in green and gray, lower and higher dose exposure, respectively. These can be compared with the 3D reconstructions of measured fluorescence image stacks. The two temperatures $T = 20\,°C$ and $T = 45\,°C$ are highlighted in blue and red, respectively. The beams start straight at $T = 20\,°C$ and are curved at $T = 45\,°C$. **b** Curvature, i.e., inverse radius obtained by fitting a circle to the experimental data, versus temperature. The right-hand side panel shows the result of twelve temperature cycles without deterioration (error bars are s.d.). **c** Bright-field optical micrographs of a $3 \times 3$ array of nominally identical structures to demonstrate the reproducibility. **d** Temperature dependence of five structures with different beam lengths prepared under identical fabrication conditions. Scale bars are $20\,\mu m$ in **a** and **b** and $50\,\mu m$ in **c** and **d**

To achieve a defined and reliable actuation, a robust fabrication process is mandatory. Therefore, we investigated the reproducibility of the production procedure and the reversibility of the actuation pattern over multiple cycles of stimulation. Figure 3c shows a top view on a substrate with several nominally identical microstructures that correspond to a design where the left bar shows a stronger thermo-response than the right one. Upon increasing the temperature to $45\,°C$ ($T > \text{LCST}$) all of the microstructures responded in an identical manner, which validates the robustness and control of the fabrication process (Supplementary Movie 2).

Applications which rely on a defined actuation behavior, e.g., in microfluidics, demand maximum control on the response. It is therefore mandatory to analyze the effect of different factors on the actuation behavior. In addition to the material properties, the amplitude of the actuation is mainly influenced by the geometry, particularly the aspect ratio of the bi-material beam[35]. Consequently, we analyzed the influence of the beam length on the response of the structures. Figure 3d depicts the reversible actuation of structures with beam lengths ($L$) from $30\,\mu m$ to $60\,\mu m$ and an extreme case of $L = 120\,\mu m$. These long beams are initially bent towards the side of the more crosslinked material due to the change of solvent from ethylene glycol to water during the development. By increasing the temperature to $45\,°C$, the beams showed large-amplitude actuation, which is completely reversible upon cooling the sample back to ambient temperature. In the case of $50\,\mu m$ beam length, the bending is already sufficiently large to touch the central pillar, whereas for $L = 120\,\mu m$ the beam is even

convolving several times to a spiral (Supplementary Movies 3, 4). The fabrication of free-hanging structures of this length without additional support is possible due to the hydrogel characteristics of the material. Since the hydrogel contains a high percentage of water, the density of the microstructure and surrounding media are similar and the beam swims and does not bend downwards due to gravity. As indicated by the arrows, this response was completely reversible. By decreasing the temperature back to $20\,°C$ ($T < \text{LCST}$) the less crosslinked material starts to swell and the beams revert back to their initial position. To confirm the reversibility of the actuation process, we performed more than ten heating/cooling cycles on the sample and no changes in the optical images recorded after each step were detected. As an additional quantification, we manually fitted a circle to the optical micrographs for the respective temperatures. This procedure was repeated three times per image. The resulting mean value of the curvature, the inverse of the circle radius, and the corresponding standard deviation of the analysis (see error bars) are depicted in Fig. 3b.

To demonstrate the large variety of possibilities with this approach we designed a micro-gripper that reversibly opens and closes on demand (see Supplementary Fig. 2).

**Complex actuation patterns of segmented microstructures**. The simple bi-material beams and grippers exhibited a controllable and large-amplitude actuation. However, depending on the application, more complex actuation patterns are required. Thus,

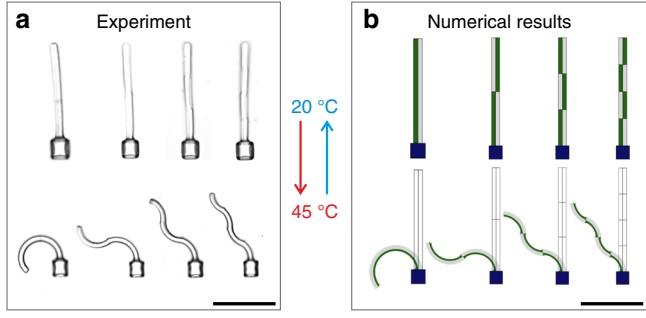

**Fig. 4** Hetero-structures with increasing complexity (from left to right). **a** Measured bright-field optical micrographs. **b** Results of corresponding numerical calculations. For both **a** and **b**, the structures on the left are simple bi-material beams (compare Fig. 3). In the structures towards the right, the two thermoelastic materials alternate $N = 2, 3, 4$ times, respectively. This leads to increasingly complex actuation results upon temperature change. Scale bars are 50 μm

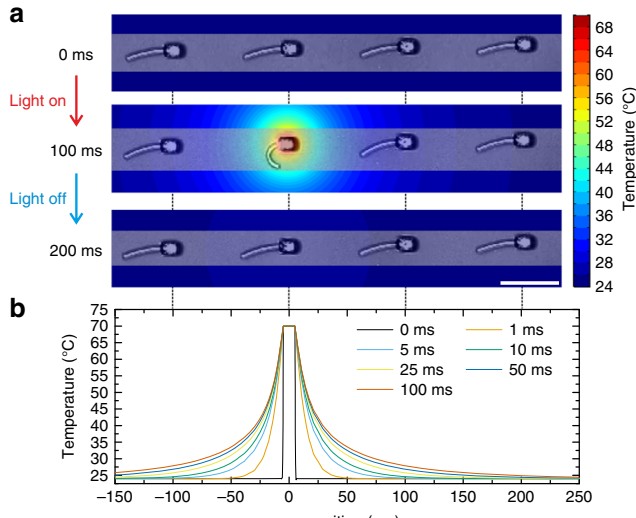

**Fig. 5** Light-induced actuation. **a** Upon focusing a femtosecond laser on one of the bases, the corresponding bi-material beam bends strongly, whereas the adjacent structures show only a weak response. Upon switching off the light, the structure moves back to its original position. This behavior is repeatable. A calculated temperature profile (see false-color scale on the right) is overlaid onto the measured optical micrographs. **b** Cross-sections through such temperature profiles in the water bath. At $t = 0$ a constant temperature is assumed for the base, whereas the outside stays at ambient conditions. With increasing time (see legend) heat diffuses away from the base, however, the temperature rise at the adjacent structures is too small for significant actuation (compare Fig. 3b). Scale bar is 50 μm

we realized hetero-structured beams which transform into different designed shapes as a response to the external stimulus. To achieve the desired behavior, we exploited the flexibility of our fabrication method. By varying the local exposure dose at different positions of the structure, we could tune the material parameters in different regions on demand. Thus, the locally induced bending of the beam can be controlled in amplitude and direction. In addition, numerical calculations to model the experimental data and predict the actuation on the basis of the design were performed.

Initially, we analyzed structures with more than two material domains to perform more sophisticated actuations. Figure 4a shows experimental data of four examples with segments of varying exposure dose along the beam. The material properties are color-coded in the numerical results in Fig. 4b. Green indicates a low exposure dose and thus a strong response, and gray a high exposure dose and thus a weak response to the stimulus. The first structure on the left-hand side is designed as a simple bi-material similar to the results shown above. The next three beams consist of two, three, and four sections, respectively, in which the dose between the left and right beam was exchanged after each part. By increasing the temperature to 45 °C, the different actuation patterns become apparent. In the case of two opposed segments, the lower half exhibits a bending to the left side, whereas the upper part bends to the right side. As a result, an s-shaped structure was formed. For more than two sections, the corresponding behavior led to even more complex shapes (compare to Supplementary Movies 5, 6). These examples demonstrate that the response to the stimulus can be precisely controlled by local variation of the exposure dose in a hetero-structure composed of two largely different ingredients.

It is desirable to develop a simple theoretical model that is capable of predicting the actuation behavior of hetero-structures. While advanced theoretical work on pNIPAM gels has been published[36,37], little is known on the mechanical properties of the materials emerging from the photoresist system investigated here, fabricated using different exposure doses. Therefore, we employed the simplest possible continuum-mechanical description, namely linear Cauchy elasticity. Importantly, geometrical nonlinearities have been accounted for in the solver. In the spirit of reverse engineering, the input parameters were chosen such that the calculations obtained for the simplest bi-material beam (left structure in Fig. 4b) closely resembled the experimental situation (left structure in Fig. 4a). Details of the used numerical approach are given in the methods section. We found that the thermal expansion coefficient changes by one order of magnitude when

increasing the exposure dose from 30 mW to 37.5 mW. For the same exposure dose variation, the Young's modulus changes by one order of magnitude, too (see methods section). Next, we applied the same equations and the same material parameters to model the three more complex structures shown in Fig. 4. The comparison of all structures in Fig. 4a, b reveals excellent agreement. Thus, the experimental findings are consistent with the simple theoretical model, hence there is no need for more advanced theoretical models. Our simple theoretical model even has predictive strength. However, our description is obviously not able to explain the molecular origin of the variation of the mechanical material properties as a function of the exposure dose. Such understanding is highly desirable to design and synthesize further new stimuli-responsive hydrogel systems, but is way beyond the scope of our paper.

**Local stimulation induced by two-photon absorption.** In all results shown so far, the stimulus was applied by globally changing the temperature. This approach is especially relevant if a homogeneous and controlled actuation over the entire substrate is required. In addition, by adjusting the geometry and the material properties, it is possible to control how the structures respond. For other applications, however, a local stimulation is desirable to only trigger the actuation at defined points of the structure. Furthermore, the timescale of a global temperature change is in the order of seconds. Often, faster switching is required. Therefore, we employed the two-photon absorption of focused light to locally increase the temperature via photo-thermal conversion[38]. These experiments were possible by employing the same set-up used for the fabrication of the samples. Figure 5 depicts the response of several hetero-microstructures with a distance of 100 μm between them before and after the laser is tightly focused onto one of them. After 100 ms of illumination, the central beam

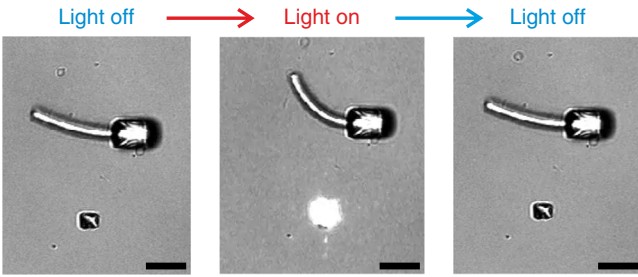

**Fig. 6** Indirect stimulation by a spatially separated structure. The bi-material beam in the top part of the images is fabricated by the pNIPAM resist as described previously, while the small blocks on the bottom are made out of PETA with the photoinitiator Irgacure 819. The laser is focused on the PETA block and by photo-thermal conversion of the two-photon absorbed light, heat is generated. This heat is distributed in the water and leads to an actuation of the bi-material beam. The PETA block itself does not experience any measureable structural change or response to the stimulation. Scale bars are 20 μm

already exhibited a large-amplitude bending, similar to the previous results obtained by increasing the temperature. For the adjacent structures, only a small bending can be observed, whereas the beam in a distance of 200 μm on the right side shows barely any response. By turning the light off, the beams moved back to their initial state after another 100 ms due to the cooling down of the irradiated region (see Supplementary Movie 7). These findings revealed that a fast and local actuation by stimulation via focused light is possible, opening further opportunities. Using a two-photon process effectively squares the intensity profile of the laser focus and thereby strongly suppresses long intensity tails, especially in the axial direction. Thus, two-photon absorption allows to concentrate the excitation in all three spatial directions in 3D.

In addition, we have performed the following control experiment: We have fabricated PETA blocks containing a two-photon initiator (see methods section) next to the stimuli-responsive pNIPAM structures (see Fig. 6). When the same femtosecond laser pulses as in the preceding paragraph were focused onto the PETA blocks, the pNIPAM structures again bended, whereas the PETA blocks exhibited no measureable response. This behavior is consistent with our interpretation that the bending of the pNIPAM structures is temperature-induced rather than being of structural origin. In yet additional control experiments, we have fabricated similar PETA blocks (not depicted), yet without a two-photon initiator (which is possible by directly exciting the HOMO-LUMO transition of the resist, albeit with reduced efficiency). We found negligible bending of the pNIPAM structures under otherwise similar conditions. We conclude, that the temperature increase is induced by two-photon absorption of the remaining photoinitiator (in this case Irgacure 819). Absorption spectra of the used photoinitiators can be found in literature[30,39,40].

However, only based on the experimental data it is challenging to extract reliable quantitative information about the temperature distribution on the substrate. Consequently, we performed a numerical analysis on the 3D heat conduction on the sample to establish a better understanding of the underlying effects. To address this specific problem in the calculations, we fixed the temperature of the central pillar of the structures. This temperature was chosen to be consistent with the movement of the beam connected to it. Around the structure, we placed a large water bath to mimic the experimental situation. Subsequently, we evaluated the temperature change as a function of time and space, taking into account heat conduction in the aqueous surrounding.

The results of these calculations are depicted in the color map and the x-line-scan through the focal spot (Fig. 5b). It was found that in the region around the illuminated pillar, where the bending of the central bar takes place, the temperature is in the range of 40 °C to 45 °C and thus, above the LCST of the material. Consequently, a strong response of the material is expected, which is in good agreement with the experimental results. Adjacent structures which are placed 100 μm away were also affected and experienced a temperature increase from 24 °C (room temperature) to about 32 °C and thus, a comparably small actuation. At 200 μm distance, the change was on the order of 1–2 °C and hence, well below the non-linear response regime of the material (see Fig. 3b).

Additionally, we used the numerical calculations to analyze the time constant of the stimulation via focused light. It is important to note that due to the nonlinear transition of the material around the LCST, the majority of the response takes place between 28 °C and 40 °C. On this basis, we estimated the time for the structures to get back in their initial state after turning off the laser beam to be in the range of 50–100 ms (see Supplementary Fig. 3 for more details). This result is in good agreement with the experiments with the respective pulsing frequencies (see Supplementary Movie 8). Overall, the study of heat transfer via numerical analysis provides a powerful tool to interpret the experimental results and to obtain a better understanding of the architectures.

## Discussion

In conclusion, we have introduced pNIPAM based 3D hetero-microstructures by two-photon laser lithography. We apply a gray-tone lithography approach to locally control the material parameters in a single production step. Thus, we realized active structures that exhibit a large-amplitude response to changes in temperature. By utilizing the flexibility of our method, we additionally created complex actuation patterns, which are in very good agreement with numerical predictions. Furthermore, we showed that the response of the structures can be activated either by globally changing the water temperature or by locally illuminating the desired microstructure with a laser focus.

## Methods

**Materials**. N-isopropylacrylamide (Sigma-Aldrich, >97%), N,N′-methylenebis (acrylamide) (Sigma-Aldrich, >99%), lithium phenyl(2,4,6-trimethylbenzoyl) phosphinate (TCI Tokio Chemical Industry, >98%), ethylene glycol (Sigma-Aldrich, >99%), acryloxyethyl thiocarbamoyl Rhodamine B (Polysciences), 3-(tri-methoxysilyl)propyl methacrylate (Sigma-Aldrich, >97%), pentaerythritol triacrylate (Sigma-Aldrich, >97%), Irgacure 819 (BASF), 7-diethylamino-3-thenoylcoumarin (J&K Scientific, 97%), acetone (Roth, > 99,5%), isopropanol (Roth, >99,5%), methyl isobutyl keton (Roth, >99%). All chemicals and solvents were used as received without further purification.

**Resist formulation**. pNIPAM-Resist: The resist was prepared dissolving 400 mg N-isopropylacrylamide, 40 mg N,N′-methylenebis(acrylamide) and 10 mg lithium phenyl(2,4,6-trimethylbenzoyl)phosphinate in 450 μL of ethylene glycol. 4 mg of acryloxyethyl thiocarbamoyl Rhodamine B were added for fluorescence imaging.

PETA-Resist: The resist was prepared by dissolving 20 mg Irgacure 819 in 980 mg of pentaerythritol triacrylate. Irgacure 819 was replaced by 7-diethylamino-3-thenoylcoumarin (DETC) for fluorescence imaging. DETC is an efficient photoinitiator for two-photon polymerization which also exhibits a strong fluorescence. This allowed us to record 3D image stacks of the fabricated PETA structures via LSM.

**Fabrication of 3D microstructures**. A commercial Direct Laser Writing setup (Photonic Professional GT, Nanoscribe GmbH) with a 25 ×, NA = 0.8 oil immersion objective was used for fabrication. To increase the adhesion of the microstructures to the glass substrate we treated plasma-cleaned coverslips with 3-(trimethoxysilyl)propyl methacrylate (1 mM in toluene) for one hour and rinsed them afterwards in acetone and water.

Valve structures: The fabrication of the pNIPAM valves inside of PETA microchannels consisted of two consecutive steps. First, the PETA tubes, along with several alignment markers, were written on the cover glass and subsequently

developed in a 1:1 mixture of methyl isobutyl keton (MIBK) and isopropanol. Second, after drying, the pNIPAM-resist was drop cast onto the structures. The mentioned markers were crucial to ensure a precise lateral positioning of the valves inside the microchannels. The positioning along the third dimension was accomplished by using the built-in interface-finder. Finally, the structures were washed with acetone and developed in water.

Bi-material beam structures: Unless stated otherwise, the less (more) crosslinked beam was fabricated with a laser power at the back focal plane of 30 mW (37.5 mW), respectively, and the post with 40 mW. The scanning speed was constant at 1 mm/s. After writing, the structures were rinsed with acetone and subsequently transferred into water for further development and storage. No post-curing treatment was applied.

**Characterization**. The mechanical analysis was performed by an atomic force microscope (NanoWizard, JPK Instruments). The included heater was used to change and control the temperature during the experiments. To avoid strong perturbations due to the bimetallic bending of the cantilever, we used uncoated MLCT cantilevers (MLCT-UC, Bruker). For the experiment shown in this work, the cantilever had a nominal spring constant of 0.03 N/m.

To analyze the response of the samples to changes in temperature we employed a confocal laser scanning fluorescent microscope (LSM 510 Meta, Zeiss), typically equipped with a 40×, NA = 0.75 air objective. To mount our coverslip, a chamber with a built-in Peltier element which is driven by an external controller was constructed. This allowed to actively heat and cool the sample while imaging the microstructures in an aqueous environment at the desired temperatures. The light activation via two-photon photo-conversion was performed in the same setup from Nanoscribe used for the fabrication.

**Numerical analysis**. The numerical analysis was performed by a finite-element approach using the commercial software COMSOL Multiphysics to solve the linear elastic Cauchy continuum-mechanics equations. Geometrical nonlinearities have been accounted for. For the bending calculations, the response of the structures to changes in temperature was introduced as a volumetric stress. The parameters used to model the two constituent materials A and B were chosen in such a way that the numerical results closely resembled the experimental situation in the simplest bi-material case. In this way, we obtained a Young's modulus of $E_A = 1$ kPa and $E_B = 11$ kPa and a thermal expansion coefficient $\alpha_A = -1 \times 10^{-2}$ 1/K and $\alpha_B = -1 \times 10^{-3}$ 1/K for the beams with lower and higher exposure dose, respectively. The Poisson's ratio was set to $\nu_{A,B} = 0.4$ in both cases. All further calculations with more complex geometries were performed with this set of parameters. In each case, the model of the structures for the calculations was identical to the respective design for the fabrication. For the heat diffusion calculations, a central base structure with a sufficiently large water bath around it was chosen. The base was set to a constant temperature to model the heating by photo-thermal conversion of the focused laser light and the outer boundaries of the water bath were set to room temperature. The discretization was chosen in an adaptive manner from a high level of detail around the base to lower level of detail at the outer boundaries.

## Data availability
The data that support the findings of this study are available within the article and its Supplementary Information file, and from the corresponding authors upon reasonable request.

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

## Acknowledgements

We acknowledge support by the Helmholtz Association via the programs "Science and Technology of Nanosystems" (STN) and "BioInterfaces in Technology and Medicine" (BIFTM), the Excellence Cluster "3D Matter Made to Order" (3DMM2O), the Nanostructure Service Laboratory (NSL) of KIT, and the Karlsruhe School of Optics and Photonics (KSOP). C.B.-K. acknowledges a Laureate Fellowship from the Australian Research Council (ARC) enabling his photochemical research program, underpinned by key support from the Queensland University of Technology (QUT).

## Author contributions

M.H., E.B., M.T., M.W., and M.B. conceived and initiated the study. M.H. performed the DLW experiments and studies of the stimuli-response. M.H. and J.Q. carried out the numerical analysis. E.B., C.B.-K., M.W., and M.B. motivated and supervised the research program. M.H., E.B., M.W., and M.B. wrote a first draft of the manuscript. All authors discussed the results and worked on the manuscript.

## Additional information

**Competing interests:** The authors declare no competing interests.

