## [Peer Review File · Nature Communications]

Reviewers' comments:

Reviewer #1 (Remarks to the Author):

Manuscript ID: NCOMMS-18-24322

Authors: Hippler et al.

The manuscript includes some major defects. In this respect, the comments are listed as follows:

Novelty issues:

1) The novelty of the research is under question. I) the responsive photoresist used in this research is well-known and was initially introduced in Ref. [24]. II) the bi-material beam presented in Fig. 1 was initially introduced by Timoshenko as a well-known method to make thermal self-bending elements. III) the hetero-structure presented in Fig. 2 was also introduced recently by Bodaghi et al. [a] (see Fig. 1 in that paper). IV) simulations by COMSOL Multiphysics were not complicated. The authors are advised to clarify them and justify the novelty of the research work.

[a] Self expanding/shrinking structures by 4D printing, Smart Mater. Struct. 25 105034, 2016.

Technical issues:

- 2) The bi-material beam was designed without taking into account their glassy transition temperatures. See Ref. [a] for more details on it.
- 3) It is needed to present thermal behaviors of the polymers by conducting DMA tests.
- 4) The results in Fig. 1b show that the curvature peak changes cycle by cycle. It should be clarified.
- 5) Young's modulus and thermal expansion coefficient were obtained by fitting experimental data via COMSOL Multiphysics. It is not a right way to calibrate them. The authors should fabricate dog-bone shape samples and conduct uniaxial tensile and thermal tests for extracting material properties.
- 6) Polymers show a combination of glassy and rubbery phases through heating and cooling. It is necessary to consider polymer phase transformation in simulation by COMSOL Multiphysics. It can be easily achieved by importing results from DMA test into the software.
- 7) The sample size is in micro level. It is necessary to consider Couple Stress Theory to simulate structural bending in this size.
- 8) Light-induced actuation presented in Fig. 3 is very fast, e.g., 100 ms. It is believed that the simulation should have been performed in a dynamic manner than a static one. It is a transient temperature-dependent deformation indeed.
- 9) Considering the content of the manuscript, the following title is suggested:
"Self-Folding 3D Microstructures Induced by Temperature and Light".

Reviewer #2 (Remarks to the Author):

This paper by Wegener, Bastmeyer, and co-workers described an interesting approach to fabricate heterogeneous hydrogel structures by means of multiphoton lithography. The hydrogel is made of thermo-responsive poly(N-isopropylacrylamide) (pNIPAM) with a lower critical solution temperature (LCST) around 35 °C. The authors tuned the local laser doses during 3D microfabrication and generated hetero-structures with variations in mechanical properties. Finally, they demonstrated two-photon-induced actuation of individual structures. In general, this paper provides an very interesting engineering approach for readers in the fields of soft robotics. While the characterization of actuation and correlation with numerical calculations are well done, a few important issues should be addressed with additional data before it is considered for publication.

Major points:

- 1) Significance and novelty of this work: There have been a few exceptional studies on pNIPAM-based or protein-based hydrogels fabricated by 2PP for soft robotics and micro-devices in previous reports (e.g. 10.1002/adma.201604825; 10.1002/adfm.201203880; 10.1073/pnas.0709571105). It is important for the authors to describe further about the novelty of this research when compared to the state-of-the-art. What are the advantages of pNIPAM in comparison to biocompatible proteins such as BSA? What are the foreseeable applications? Are there additional novelty besides so called „more complex actuation,,?
- 2) Detailed procedure for laser microfabrication: It will be very helpful if the authors could provide more details about the microfabrication procedure. How much is the specific laser doses in the two structural parts? What are the minimal difference in term of laser processing parameters to ensure that the cylindrical part remains stable, the other part is motile?
- 3) Additional materials characterization: The authors reports a recipe for making the pNIPAM gels. What are the know-hows behind figuring out this „optimal,, recipe? I.e., how does the molar concentration of crosslinker influence gel properties and their other features (swelling, thermal response)? How does laser intensity influences their crosslinking density, temperature-dependent (de-)swelling and thermal response?
- 4) A thorough mechanical analysis of the microstructure (laser-dose-dependent) is needed to support the discussion on actuation.
- 5) Ethylene glycol is used to make the formulations. Is it possible to use water or PBS instead? What are the minimal polymer concentration for two-photon polymerization?
- 6) Figure 3-light-induced actuation: It would be helpful if the authors can show that the actuation is exclusively controlled through a described mechanism, not by other factors. Control experiments using non-pNIPAM materials may help test the hypothesis. Is the actuation partially due to laser-induced additional crosslinking or even de-crosslinking (degradation) if too much dose? If so what is the threshold laser dose for a non-destructive actuation?
- 7) The authors claimed „complex actuation,, several times. But the results and videos do not show a rather complicated actuation apart from twisting motions. It would be important to show more complex actuations such as 3D-shape origami to support this claim.

Reviewer #3 (Remarks to the Author):

"Controlling the Shape of 3D Microstructures by Temperature and Light":

The manuscript presents 3D printed hetero-microstructures composed of poly(N-isopropylacrylamide) (pNIPAM). The authors present bi-layer micro-actuators fabricated from a single photoresist, where the difference between the two layers is the amount of exposure, hence, degree of crosslinking.

The authors optimized the composition of the photoresist, investigated the thermal-response of the actuator, including the effect of the actuator's length, and showed reversibility and reproducibility of the actuation.

In addition, the authors fabricated actuators with complex actuation patterns using local exposure and predicted the actuation with numerical calculations.

Finally, the authors demonstrated actuation using focused light, achieving local movement and fast millisecond response.

The report presents interesting and novel approach for fabrication of bi-layer micro-actuator and is suitable for publication in the journal. However it presents a possibility for an actuator, but does not present a real device. If this is added to the results, the paper would have a much higher impact.

In addition, the following comments should be addressed before publication:

Results and discussion:

- "When the temperature is only slightly elevated above its LCST (page 3 line 75)"- LCST should

be written in full when first written in the main text (abstract not included).

- "...during gray-tone lithography (page 3 line 85)"- a reference or an explanation of the method is required.
- "lithium phenyl-2,4,6-trimethylbenzoylphosphinate,24 : thus reference is related to other photoinitiators , so it should be replaced.
- "The results indicated that the material transition from hydrophilic to hydrophobic is not as sharp as in the case of the non-crosslinked material.25 (page 6 lines 143-144)"- It is unclear how this statement was deduced from the graph. Which parameter was compared?
- "...the amplitude of the actuation is mainly influenced by the geometry, particularly the aspect ratio of the bi-material beam. (page 7 line 159-160)"- This statement should be supported by a reference or theory.
- Figure 1d- the different length should be stated on the figure itself.
- Figure 1d- the initial position of the beams in this figure is a bit curved to the right-hand side (towards the material with higher crosslinking density), why is that? An explanation is needed.
- "By decreasing the temperature back to 20 °C ($T < LCTS$), the crosslinked material starts to swell and the beams revert back to their initial position (page 7 lines 171-172)"- shouldn't it be the less crosslinked material?
- "Therefore, we employed the two-photon absorption of focused light to locally increase the temperature via photo-thermal conversion (page 11 lines 238-239)"- a reference is required.
- "...well below the non-linear response regime of the material (see figure 2b). (page 12 lines 266-267)"- should be figure 3b.
- "Since a two-photon process is involved in the absorption, even a local addressing in 3D might be possible. (page 11 lines 248-250)"- the sentence is unclear.

Methods:

- "After writing, the structures were rinsed with acetone and subsequently transferred in water for further development and storage (page 13 lines 303-304)"- was there any post-curing treatment? If there was it needs to be detailed.
- "...we obtained a Young's modulus of... (page 14 line 317)"- How was this determined ? Are the values for Young's modulus and thermal expansion coefficient based on measurements? literature? It should be explained.

Supporting information:

- The numbering of the movies is incorrect, movie M7 is missing.

In what follows, *we repeat the comments of the reviewers in red and italic*, respond to them in green, repeat passages from the original manuscript in black, and highlight changes made in the revised version of our manuscript in blue.

Reviewer #1

The manuscript includes some major defects. In this respect, the comments are listed as follows:

Novelty issues:

1) The novelty of the research is under question. I) the responsive photoresist used in this research is well-known and was initially introduced in Ref. [24]. II) the bi-material beam presented in Fig. 1 was initially introduced by Timoshenko as a well-known method to make thermal self-bending elements. III) the hetero-structure presented in Fig. 2 was also introduced recently by Bodaghi et al. [a] (see Fig. 1 in that paper). IV) simulations by COMSOL Multiphysics were not complicated. The authors are advised to clarify them and justify the novelty of the research work.

[a] Self expanding/shrinking structures by 4D printing, Smart Mater. Struct. 25 105034, 2016.

I) Indeed, it is well known and cited as such “well known” system in our paper.

II) Bi-materials beams are so well known that one usually does not even give a reference anymore.

III) Figure 1 of reference [a] is merely a scheme, not a fabricated structure. The structures that are presented later in that paper are macroscopic (and not microscopic as ours). There are no measurements anywhere close to the ones we present in our paper.

IV) We make no claim that the COMSOL calculations are complicated. This aspect is beside the point. We just use these calculations to retrieve effective parameters (also see below). Our experiments would even speak without any such modelling.

In brief, as explained in the introduction of our paper, the novelty of our work lies in that we start from a *single* resist formulation, which is then exposed by different laser powers to lead to 3D *microstructures* composed of *two materials with drastically different properties* (“gray-tone lithography”). The second novelty of the present work is that a *local* temperature increase and hence the actuation, can be induced by *two-photon* absorption of focused light. This aspect potentially allows for initiating local responses in *three* dimensions, *i.e.*, not just on surfaces but also *inside* of 3D structures.

It is interesting to note that the first sentence of the report of reviewer #2 grasps this novelty right away.

We will come back to the aspect of novelty in our response to point 1) of reviewer #2 (see below).

Technical Issues:

2) The bi-material beam was designed without taking into account their glassy transition temperatures. See Ref. [a] for more details on it.

We note to the reviewer that our system is based on pNIPAM, whose thermo-response relies on a sharp transition temperature in water, at which the material becomes either hydrophobic or hydrophilic resulting in a shape change (shrinking/swelling). In sharp contrast, the system

reported in reference [a] is based on macroscopic bulk materials composed of shape memory polymers, whose self-expanding/shrinking mechanism relies on a glass transition. Thus, our system did not require the optimization of its glass transition temperature (T_g), but the lower critical solution temperature (LSCT). Indeed, the LSCT was taken in account by selecting pNIPAM as the thermos-responsive material (see introduction of the manuscript) and further developed to achieve more complex actuation by creating hetero-microstructures.

3) It is needed to present thermal behaviors of the polymers by conducting DMA tests.

We tune the material parameters originating from just a single resist by laser power. When writing a beam with large laser power and then an adjacent one with low laser power, the former is influenced by the latter. Therefore, one can NOT simply analyze the two materials one by one. Furthermore, the reviewer may have overlooked that all measurements have to be performed under aqueous conditions and that the sample should never leave the aqueous environment.

4) The results in Fig. 1b show that the curvature peak changes cycle by cycle. It should be clarified.

When looking at the right part of Figure 1b (Figure 3b, according to the new numbering), it becomes clear that the first and the last data point agree within error bars. The data points fluctuate a bit but there is no evidence for irreversible deterioration. In fact, we have looked at even much longer time series supporting our claim. For example, movie S8 shows more than 50 cycles with different exposure times on the same structure.

5) Young's modulus and thermal expansion coefficient were obtained by fitting experimental data via COMSOL Multiphysics. It is not a right way to calibrate them. The authors should fabricate dog-bone shape samples and conduct uniaxial tensile and thermal tests for extracting material properties.

We respectfully disagree. As already explained in point 3) above, one cannot inspect the two materials separately. Again, the experiments suggested by the reviewer are incompatible with the requirement that the samples need to stay in aqueous conditions at all times.

6) Polymers show a combination of glassy and rubbery phases through heating and cooling. It is necessary to consider polymer phase transformation in simulation by COMSOL Multiphysics. It can be easily achieved by importing results from DMA test into the software.

Following points 2) and 3), point 6) has become obsolete. Furthermore, it is not necessary to consider any phase transition in the calculations. What the calculations essentially do is to consider only the end points of the motion, *i.e.*, one lower temperature and one higher temperature. All that enters is the volume at the two temperatures, the Young's moduli, and the thermal expansion coefficients at these temperatures. The underlying mechanism is strictly irrelevant in these calculations

7) The sample size is in micro level. It is necessary to consider Couple Stress Theory to simulate structural bending in this size.

Many papers on mechanical metamaterials in the same size regime (for example the December 2017 Science paper of the Wegener group [b]) have shown that the polymer itself is within the scalable regime. In this regime, our calculations account for all aspects covered by Cauchy elasticity, including the coupling between two constituents in the bi-material beams. Nothing is neglected.

[b] Frenzel, T., Kadic, M., & Wegener, M. Three-dimensional mechanical metamaterials with a twist. *Science*, **358**(6366), 1072-1074 (2017).

8) Light-induced actuation presented in Fig. 3 is very fast, e.g., 100 ms. It is believed that the simulation should have been performed in a dynamic manner than a static one. It is a transient temperature-dependent deformation indeed.

The reviewer may have overlooked that Figure 3b (numbering according to the original manuscript, Figure 5b in the numbering of the revised version) shows the requested dynamic calculations.

9) Considering the content of the manuscript, the following title is suggested: "Self-Folding 3D Microstructures Induced by Temperature and Light".

We respectfully disagree. We do not only show "folding" but also "wrinkling" in Figure 2. Furthermore, in response to point 1) of reviewer #2, we have added two new figures with additional data and example geometries. On this basis, the suggested title would be insufficient.

Reviewer #2

This paper by Wegener, Bastmeyer, and co-workers described an interesting approach to fabricate heterogeneous hydrogel structures by means of multiphoton lithography. The hydrogel is made of thermo-responsive poly(N-isopropylacrylamide) (pNIPAM) with a lower critical solution temperature (LCST) around 35 °C. The authors tuned the local laser doses during 3D microfabrication and generated hetero-structures with variations in mechanical properties. Finally, they demonstrated two-photon-induced actuation of individual structures. In general, this paper provides an very interesting engineering approach for readers in the fields of soft robotics. While the characterization of actuation and correlation with numerical calculations are well done, a few important issues should be addressed with additional data before it is considered for publication.

We thank the reviewer for this overall very positive feedback to our work.

Major points:

1) Significance and novelty of this work: There have been a few exceptional studies on pNIPAM-based or protein-based hydrogels fabricated by 2PP for soft robotics and micro-devices in previous reports (e.g. 10.1002/adma.201604825; 10.1002/adfm.201203880; 10.1073/pnas.0709571105). It is important for the authors to describe further about the novelty of this research when compared to the state-of-the-art. What are the advantages of pNIPAM in comparison to biocompatible proteins such as BSA?

The first paper (10.1002/adma.201604825) is an excellent study using pNIPAM based structures. However, the authors do not use a 3D fabrication approach. They rather fabricate stripes by 2D molding. Thereafter, the stripes are coated with gold. Helices form by spontaneous symmetry breaking. Clearly, our 3D structures (see examples shown in Figures 2,3 and 4) cannot be fabricated along these lines. This aspect represents a novelty of our work.

The second publication (10.1002/adfm.201203880) is a feature article that mainly refers to the third one in regards to stimuli-responsive aspects.

The third paper (10.1073/pnas.0709571105) is a nice work using the protein BSA. However, the stimulus is a drastic change of the pH value in the range between 2 and 14. These extreme values are not biocompatible. In our work, a strong response can already be triggered by only slightly changing the temperature from 37°C (culture conditions) to e.g. 32°C. Therefore, importantly, our resist system is biocompatible. Furthermore, our resist system allows for local stimulation by light. In sharp contrast, changing the pH level results in a global stimulus. The combination of these aspects forms another novelty of our work.

To further describe the novelty of our research we rewrote the last paragraph of the introduction:

“...In this paper, we introduce a composite resist formulation based on *N*-isopropylacrylamide and the crosslinker *N,N'*-methylenebisacrylamide. The novelty of this composite is twofold. First, the local properties in a three-dimensional (3D) microstructure can be tuned by the local exposure dose during 3D laser lithography (gray-tone lithography), opening the door to 3D hetero-microstructures from a single resist. The resulting property differences are extremely large. Our experimental results are consistent with numerical calculations which indicate a ten-fold change in the thermal expansion coefficient and the Young’s modulus versus temperature during gray-tone lithography. As examples, we demonstrate a variety of complex 3D architectures exhibiting large-amplitude and complex actuation responses. The second novelty of the present work is that a local temperature increase and hence the actuation can be induced by two-photon absorption of focused light. This aspect potentially allows for initiating local responses in three dimensions, *i.e.*, not just on surfaces but also inside of 3D structures....”

What are the foreseeable applications?

- Actuation and gripping for soft robotics applications. To illustrate this aspect, we have added the data shown in Figure S2

Supplement Figure S2. Operation of a bi-material gripper. The gripper consists of two bi-material beams with the less crosslinked part at the outer side. In this configuration, the gripper is closed at room temperature and opened at elevated temperature.

with a corresponding reference in the text:

“...As an additional quantification, we determined the curvature of the beams in the two states and observed no significant deviation over the cycles (Figure 3b). To demonstrate the large variety of possibilities with this approach we designed a micro-gripper that reversibly opens and closes on demand (see supplement Figure S2)...”

- Valve systems in 3D microfluidic channels. To illustrate this aspect, we have added the data shown in Figure 2

Figure 2. Stimuli-responsive pNIPAM valves (colored in gray) in a non-responsive microchannel made of pentaerythritol triacrylate (PETA, colored in turquoise). a) 3D reconstruction of experimental data recorded via confocal laser scanning microscopy. Upon heating to the sample to 45 °C the valve shrinks and the channel widens. This process takes place reversibly when the sample is cooled back down. b) Open area in the channel at 20 °C and 50 °C for multiple cycles of stimulation with no significant deviation. c) Alternative design with an additional inner tube and two pNIPAM-tori. Complete closure of the channel can be achieved in a reversible manner.

and a corresponding section in the main text:

“...3D laser lithography is a versatile technique that readily allows the combination of different photoresist systems. To highlight the potential of our resist system in this regard, we next demonstrated the combination of responsive photoresists with conventional non-responsive materials. As an example, we designed micrometer-scale rigid tubes with stimuli-responsive valves. Figure 2a shows a top view and a x,z-cut of a 3D reconstruction of experimental data, obtained by confocal laser scanning microscopy (LSM). In a first writing step, a common photoresist based on pentaerythritol triacrylate (PETA, colored in turquoise) was used to fabricate the tubes. Subsequently, a pNIPAM-based torus (colored in grey) was written inside the tube at a height of 20 μm. To record these images, we designed a chamber with a Peltier element to heat or cool the sample to the desired temperature.

The graph in Figure 2b depicts the open area of the channel for multiple cycles of stimulation. By increasing the temperature to 45°C, the hydrogel shrunk significantly and the tube channel widened. As a consequence, the measured open channel area increased by more than a factor of 2.5. Upon cooling the sample back down to room temperature the initial situation was restored. Importantly, only the pNIPAM valves reacted to the stimulus, while the structures made from PETA remain unchanged. We performed multiple cycles of heating and cooling and found no deterioration within measurement errors. An alternative design with an additional inner pNIPAM-tori is depicted in Figure 2c. At 20 °C both tori are swollen and close the channel completely. At elevated temperatures the hydrogel shrinks and the channel opens.

To achieve a yet more complex response and large-amplitude actuation, more sophisticated structures with a high spatial control of the material parameter are required. In this regard, the previously shown sequential approach is inherently limited. Thus, the

next step was the fabrication of responsive 3D hetero-microstructures from a single resist and subsequent investigation of the thermo-response....”

- Local stimulation of biological cells by focused light and two photon absorption thereof. This aspect is illustrated in Figure 5.

Are there additional novelty besides so called „more complex actuation,,?

Several aspects of novelty have been discussed in detail above. In addition, we emphasize again that our approach starts from a single resist formulation and uses the light dose to induce substantially different material properties. Furthermore, our approach allows for combining this resist system with other established photoresist platforms. This aspect is now illustrated by adding the data shown in Figure 2.

2) Detailed procedure for laser microfabrication: It will be very helpful if the authors could provide more details about the microfabrication procedure. How much is the specific laser doses in the two structural parts?

The less (more) crosslinked beam was fabricated with a laser power at the back focal plane of 30 mW (37.5 mW). The scanning speed was constant at 1 mm/s. These data are now given in the revised version of the Methods section:

“... 25x, NA=0.8 oil immersion objective was used for fabrication. Typically, the less (more) crosslinked beam was fabricated with a laser power at the back focal plane of 30 mW (37.5 mW) respectively and the post with 40 mW. The scanning speed was constant at 1 mm/s. To increase the adhesion of the microstructure...”

What are the minimal difference in term of laser processing parameters to ensure that the cylindrical part remains stable, the other part is motile?

The used laser powers and parameters are also given in the revised version of the Methods section. In brief, the ratio of laser powers used for the post and the soft part of the bi-material beam is less than a factor of 2. The fact that this ratio is so small represents one of the attractive features of our resist system.

3) Additional materials characterization: The authors reports a recipe for making the pNIPAM gels.

What are the know-hows behind figuring out this „optimal,, recipe? I.e., how does the molar concentration of crosslinker influence gel properties and their other features (swelling, thermal response)? How does laser intensity influences their crosslinking density, temperature-dependent (de-)swelling and thermal response?

These aspects are discussed in the revised version of the main text. In brief, the relative contributions of crosslinker molecules and NIPAM molecules is a trade-off. For very small crosslinker concentrations, one can hardly write 3D structures at all. The resulting material responds to the stimulus though. In the opposite limit of large crosslinker concentrations, one can easily write arbitrary 3D structures. However, these structures hardly respond to the stimulus temperature. The optimum crosslinker concentration lies between these limits. As quoted in the main text, the optimum molar ratio of NIPAM to crosslinker for the architecture discussed in our paper is 14:1. This value has been found by systematically increasing the crosslinker concentration.

The influence of the laser intensity on the crosslinking density is described in the manuscript. On page 7-8 of the revised main text we write:

“...As discussed previously, by increasing the temperature from $T=20\text{ }^{\circ}\text{C}$ to $T=45\text{ }^{\circ}\text{C}$, the LCST of the material is exceeded and the material shrinks and stiffens. The magnitude of this effect, however, is largely dependent on the crosslinking density of the polymerized hydrogel. If more fixed crosslinks between the polymer chains are present, the material is strongly confined to the geometry of the fabrication design. As a result, the material showed a much smaller thermo-response compared to a weakly crosslinked hydrogel....”

Additional data is depicted in Figure S1:

“...The relation between the exposure dose and the beam actuation was carefully assessed and hetero-microstructures were prepared by varying the fabrication parameters (Figure S1 in the supplementary information section)....”

4) A thorough mechanical analysis of the microstructure (laser-dose-dependent) is needed to support the discussion on actuation.

See discussion for reviewer #1. Mechanical analysis is only possible for “single” materials. In the situation of the bi-material beam, the writing of one bar influences the result for the one written thereafter. Therefore, we show data for different writing laser powers in Figure S1. In addition, we present an independent mechanical analysis for one single material in the new Figure 1

Figure 1. Mechanical analysis of a pNIPAM block fabricated 3D laser lithography. a) Optical micrograph in the atomic force microscope (AFM) with overlaid indications for the force measurements and the line-scan. b) Measured Young's Modulus as a function of temperature for a stepwise heating and cooling of the sample. c) Height measurement via line-scanning from the glass substrate on top of the pNIPAM block. The different colors depict several cycles of heating and cooling.

and added a corresponding section in the main text:

“...To investigate the stimuli-responsive properties of the material, we performed a temperature-dependent mechanical analysis of pNIPAM-based hydrogel blocks produced by 3D laser lithography via atomic force microscopy (AFM). Figure 1a shows an optical micrograph of such a block with the cantilever approached to the surface. To compare the behavior of our material system to commonly used macroscopic pNIPAM-based hydrogels, we evaluated the Young's modulus of the fabricated block as a function of temperature (Figure 1b). By increasing the temperature from $T=22\text{ }^{\circ}\text{C}$ to $T=43\text{ }^{\circ}\text{C}$, the lower critical solution temperature (LCST) of the

material is exceeded and the material shrinks and stiffens. As a consequence, the measured Young's modulus increased by an order of magnitude. Furthermore, the results indicated that the material transition from hydrophilic to hydrophobic is not as sharp as in the case of the non-crosslinked material.²⁶ There, the transition occurs almost exclusively in the narrow regime between 32°C and 33°C. In our case, the transition is distributed over the temperature range from 28°C and 43°C. This finding is in agreement with previous results.^{23,27,28} As a consequence, the images in this report are recorded at 20 °C and 45 °C to capture the experimental situation below and above the LCST, respectively.

The extend of swelling in this transition is visualized in Figure 1c. The height profile was recorded by performing a line-scan with the cantilever tip from the glass surface to the pNIPAM block (line indicated in Figure 1a). The results show that the hydrogel shrunk by a factor of 3 upon increasing the temperature and re-swelled to the original height after cooling the sample back down. The graph depicts three consecutive cycles of heating and cooling for the same hydrogel block. Importantly, the response of the mechanical properties and the swelling of the microstructure was completely reversible within measurement errors..."

Additionally, we included a section in the Method part to describe the characterization:

"The mechanical analysis was performed by an atomic force microscope (NanoWizard, JPK Instruments). The included heater was used to change and control the temperature during the experiments. To avoid strong perturbations due to the bimetallic bending of the cantilever, we used uncoated MLCT cantilevers (MLCT-UC, Bruker). For the experiment shown in this work, the cantilever had a nominal spring constant of 0.03 N/m."

5) Ethylene glycol is used to make the formulations. Is it possible to use water or PBS instead?

It is indeed possible to use water or PBS instead. In fact, we started with water in this study. Later, we included ethylene glycol in the formulation instead of water because ethylene glycol is a better solvent for all constituents of our resist system. Due to this better solubility, we can use a smaller percentage of ethylene glycol than water. Generally, small solvent concentrations, *i.e.*, high concentration of reactive monomers, are desirable for direct laser writing. After the development, the structures are transferred to water, leading to biocompatibility.

What are the minimal polymer concentration for two-photon polymerization?

We have not investigated this question. As discussed above, it is rather desirable to maximize the monomer concentration in the solvent to obtain mechanically stable 3D structures.

6) Figure 3-light-induced actuation: It would be helpful if the authors can show that the actuation is exclusively controlled through a described mechanism, not by other factors. Control experiments using non-pNIPAM materials may help test the hypothesis.

We have performed corresponding control experiments, which are shown in the new Figure S4.

Supplement Figure S4. Indirect stimulation by a spatially separated structure. The bi-material beam in the top part of the images is fabricated by the pNIPAM resist as described previously, while the small blocks on the bottom are made out of PETA. The laser is focused on the PETA block and by photo-thermal conversion of the two-photon absorbed light, heat is generated. This heat is distributed in the water and leads to an actuation of the bi-material beam. The PETA block itself does not experience any structural change or response to the stimulation.

Herein, we have fabricated absorbing blocks made of pentaerythritol triacrylate (PETA) next to stimuli responsive pNIPAM bi-material beams. In this situation, the heat-generating structure is spatially separated from the stimuli-responsive structure.

Additionally, we carried out control experiments with PETA blocks with and without photoinitiator respectively. The blocks with photoinitiator lead to comparable light induced heat generation and hence actuation as for the NIPAM based resist system. In sharp contrast, the blocks without photoinitiator lead to negligible light induced actuation for comparable laser powers. From these facts we conclude that the photoinitiator is the primary two-photon light absorbing component.

To clarify the fabrication, we have added the recipe for the PETA based resist in the Methods section:

“PETA-Resist: The resist was prepared by dissolving 20 mg Irgacure 819 in 980 mg of pentaerythritol triacrylate. Irgacure 819 was replaced by 7-diethylamino-3-thenoylcoumarin for fluorescence imaging.”

Is the actuation partially due to laser-induced additional crosslinking or even de-crosslinking (degradation) if too much dose? If so what is the threshold laser dose for a non-destructive actuation?

This question is connected to the previous paragraph. Consistent with this conclusion, we observe that the light actuation slowly deteriorates versus time. This process is due to gradual bleaching of the photoinitiator molecules. Laser induced additional crosslinking or even de-crosslinking can be excluded as the dominant mechanism by the data shown in the new Figure S4, where the heater is spatially separated from the bi-material microstructure.

7) The authors claimed „complex actuation,, several times. But the results and videos do not show a rather complicated actuation apart from twisting motions. It would be important to show more complex actuations such as 3D-shape origami to support this claim.

This point is closely connected to point 1) of this reviewer. In response to point 1), we have added two new figures showing other actuated geometries. All of these should be seen as

examples. On this basis, it should be clear that our approach allows for the making of a large variety of different architectures.

Reviewer #3

"Controlling the Shape of 3D Microstructures by Temperature and Light":

The manuscript presents 3D printed hetero-microstructures composed of poly(N-isopropylacrylamide) (pNIPAM). The authors present bi-layer micro-actuators fabricated from a single photoresist, where the difference between the two layers is the amount of exposure, hence, degree of crosslinking.

The authors optimized the composition of the photoresist, investigated the thermal-response of the actuator, including the effect of the actuator's length, and showed reversibility and reproducibility of the actuation.

In addition, the authors fabricated actuators with complex actuation patterns using local exposure and predicted the actuation with numerical calculations.

Finally, the authors demonstrated actuation using focused light, achieving local movement and fast millisecond response.

The report presents interesting and novel approach for fabrication of bi-layer micro-actuator and is suitable for publication in the journal. However it presents a possibility for an actuator, but does not present a real device. If this is added to the results, the paper would have a much higher impact.

We thank the reviewer for this overall very positive assessment. In response to point 1) of reviewer #2, we have added other 3D structures and have discussed potential applications. We kindly refer reviewer #3 to our response above.

In addition, the following comments should be addressed before publication:

Results and discussion:

- "When the temperature is only slightly elevated above its LCST (page 3 line 75)"- LCST should be written in full when first written in the main text (abstract not included).*

We have changed the manuscript accordingly.

- "...during gray-tone lithography (page 3 line 85)"- a reference or an explanation of the method is required.*

We have added an explanation and a reference. In the main text, we now write:

"... We exploited the flexibility of 3D laser lithography to vary the local exposure dose during writing. This gray-tone lithography approach allows us to realize materials with substantially different properties in one fabrication step from a single photoresist formulation. As usual, the notion gray-tone lithography refers to gradually changing the material properties by continuously varying the exposure dose during the process.²⁹ In this way, we achieved a highly localized control over the crosslinking density and consequently of the thermo-response of the structure..."

- "lithium phenyl-2,4,6-trimethylbenzoylphosphinate,²⁴ : thus reference is related to other photoinitiators , so it should be replaced.*

Confusingly, this reference uses a different abbreviation (Li-TPO) for the same initiator. To clarify, we have added to the main text:

“...Furthermore, the responsive photoresist contains a highly efficient photoinitiator, *i.e.* lithium phenyl-2,4,6-trimethylbenzoylphosphinate,²⁵ (abbreviated as Li-TPO in ref. [25]) and acryloxyethyl thiocarbonyl Rhodamine B as a fluorescent dye to be able to record 3D fluorescence image stacks....”

• *"The results indicated that the material transition from hydrophilic to hydrophobic is not as sharp as in the case of the non-crosslinked material.25 (page 6 lines 143-144)"- It is unclear how this statement was deduced from the graph. Which parameter was compared?*

To improve our paper in this regard we have expanded our main text:

“...The results indicated that the material transition from hydrophilic to hydrophobic is not as sharp as in the case of the non-crosslinked material.²⁶ There, the transition occurs almost exclusively in the narrow regime between 32°C and 33°C. In our case, the transition is distributed over the temperature range from 28°C and 40°C. This finding is in agreement with previous results.^{23,27,28} As a consequence, the images in this report are recorded at 20 °C and 45 °C to capture the experimental situation below and above the LCST, respectively....”

• *"...the amplitude of the actuation is mainly influenced by the geometry, particularly the aspect ratio of the bi-material beam. (page 7 line 159-160)"- This statement should be supported by a reference or theory.*

We have included a corresponding reference (R.S. Lakes, J. Mater. Sci. Lett. 15, 475 (1996)).

• *Figure 1d- the different length should be stated on the figure itself.*

The figure itself has been changed accordingly.

• *Figure 1d- the initial position of the beams in this figure is a bit curved to the right-hand side (towards the material with higher crosslinking density), why is that? An explanation is needed.*

This pre-bending is due to the change of solvent from ethylene-glycol to water. This aspect is now explained in the revised version of the main text:

“...Figure 3d depicts the reversible actuation of structures with beam lengths (L) from 30 μm to 60 μm and an extreme case of $L = 120 \mu\text{m}$. These long beams are initially bent towards the side of the more crosslinked material due to the change of solvent from ethylene glycol to water during the development. By increasing the temperature to 45 °C, the beams showed large-amplitude actuation, which is completely reversible upon cooling the sample back to ambient temperature...”

• *"By decreasing the temperature back to 20 °C ($T < LCTS$), the crosslinked material starts to swell and the beams revert back to their initial position (page 7 lines 171-172)"- shouldn't it be the less crosslinked material?*

We have corrected this typo. Indeed, it is the less crosslinked material that experiences a stronger swelling.

- *"Therefore, we employed the two-photon absorption of focused light to locally increase the temperature via photo-thermal conversion (page 11 lines 238-239)"- a reference is required.*

We are not aware of such a reference showing light induced actuation by two-photon absorption. This aspect represents a novelty of our work. We have added a loosely connected reference showing two photon photo-thermal energy conversion in the context of cancer therapy.

- *"...well below the non-linear response regime of the material (see Figure 2b). (page 12 lines 266-267)"- should be Figure 3b.*

This mistake has been corrected. Note, however, that the figure numbers have changed due to adding two completely new figures to the main text.

- *"Since a two-photon process is involved in the absorption, even a local addressing in 3D might be possible. (page 11 lines 248-250)"- the sentence is unclear.*

This aspect is perfectly analogous to the two-photon induced writing process in DLW. To improve our paper for the reader in this regard, we have changed the main text to:

"...These findings revealed that a fast and local actuation by stimulation via focused light is possible, opening further opportunities. Using a two-photon-process effectively squares the intensity profile of the laser focus and thereby strongly suppresses long intensity tails, especially in the axial direction. Thus, two-photon absorption allows to concentrate the excitation in all three spatial directions in 3D...."

Methods:

- *"After writing, the structures were rinsed with acetone and subsequently transferred in water for further development and storage (page 13 lines 303-304)"- was there any post-curing treatment? If there was it needs to be detailed.*

No post-curing treatment was applied. This is now stated in the Methods section:

"...After writing, the structures were rinsed with acetone and subsequently transferred in water for further development and storage. No post-curing treatment was applied."

- *"...we obtained a Young's modulus of... (page 14 line 317)"- How was this determined? Are the values for Young's modulus and thermal expansion coefficient based on measurements? literature? It should be explained.*

The determination of the Young's moduli and the thermal expansion coefficients are explained in the Methods section:

"...The parameters used to model the two constituent materials A and B were chosen in such a way that the numerical results closely resemble the experimental situation in the simplest bi-material case. In this way, we obtained a Young's modulus of $E_A = 1$ kPa and $E_B = 11$ kPa and a thermal expansion coefficient $\alpha_A = -1 \times 10^{-2}$ 1/K and $\alpha_B = -1 \times 10^{-3}$ 1/K for the beams with lower and higher exposure dose respectively...."

This means, that these values are not directly measured but rather retrieved by the approach describe above.

Supporting information:

- *The numbering of the movies is incorrect, movie M7 is missing.*

The numbering of the movies has been corrected.

Reviewers' comments:

Reviewer #1 (Remarks to the Author):

The comment no. 1 has not been addressed well. The introduction of the manuscript is too brief. It is needed to review more papers published in recent years on the same or related topic. For instance, the following papers are macroscopic counterpart of the current manuscript:

[a] Self expanding/shrinking structures by 4D printing, Smart Mater. Struct. 25 105034, 2016.

[b] 3D printed reversible shape changing components with stimuli responsive materials, Sci. Rep. 6 24761, 2016.

The manuscript would be recommended for publication after addressing the above mentioned comment.

Reviewer #2 (Remarks to the Author):

The authors have addressed quite a few of my comments and have improved the manuscript with additional experiments. I have the following questions/comments to add:

1. There are a few typos in the response letter and revised manuscript. E.g. p5, line 131. Please correct.
2. The authors used a few different terms as to material design, such as single resist - two materials - bi-material - composite. Altogether, these terms are quite confusing for a non-specialist. The major difference is in their physical properties, but the same chemistry. Would it be possible to improve this for a better readability?
3. Figure 3b: what are the possible reasons for the fluctuation? Is it due to inherent limitations with the material or the method to quantify curvature?
4. Limitation in simulation: It would be helpful for the authors to add an in-depth discussion with the limitations in the methodology selected for simulation.
5. Issue on Sample size: I do not think citing a Science article is sufficient to address this important point. The photoresist in the cited Metamaterial work was IP-S, which I think is totally different from pNIPAM in physical properties and mechanical response. Are there previous reports on the simulation of thermo-responsive gels like polyNIPAM? Is there non-linear elasticity with gel-like soft matter?
6. Considering the importance of photoinitiator for 2P-actuation, it would be important to include the content on this regard the main manuscript. Is the actuation efficiency dependent on the PI concentration? These control experiments are important.
7. Single-photon (or two-photon) absorption spectrum of preformed pNIPAM sample should be provided.
8. Figure 2, please indicate why the PETA part is blue-colored.
9. Page 16, line 366 in Methods section: Sentence „I819 was replaced by ...,, is unclear. Is it a mixture of PI and dye? Or this dye was used as PI?
10. Page 15, Methods section: Detailed experimental procedure for Figure 2 should be provided. It would be important for the authors to comment on the key fabrication steps.
11. Supplement Figure S4 on control experiments should be included in the SI and discussed in the manuscript.

Reviewer #3 (Remarks to the Author):

The authors have addressed properly all comments except for two:

1. The comment regarding the photo initiator reference-

The answer of the authors is incorrect, although the photo initiators are similar, Li-TPO stands for lithium 2,4,6-trimethylbenzoyldiphenyl phosphine oxide and not for lithium phenyl(2,4,6-trimethylbenzoyl)phosphinate that is used in the report.

2. The comment regarding presenting Young's modulus values-

The authors did not provided results for this property. Why did they only estimate the values ?

In what follows, *we repeat the comments of the reviewers in red and italic*, respond to them in green, repeat passages from the original manuscript in black, and highlight changes made in the revised version of our manuscript in blue.

Reviewer #1:

The comment no. 1 has not been addressed well. The introduction of the manuscript is too brief. It is needed to review more papers published in recent years on the same or related topic. For instance, the following papers are macroscopic counterpart of the current manuscript:

[a] Self expanding/shrinking structures by 4D printing, Smart Mater. Struct. 25 105034, 2016.

[b] 3D printed reversible shape changing components with stimuli responsive materials, Sci. Rep. 6 24761, 2016.

The manuscript would be recommended for publication after addressing the above mentioned comment.

Following this suggestions, we have added the two mentioned references as well as additional ones to our introduction. The references given include extensive review articles. We feel, that it would be well beyond the scope of our paper to review the entire field of stimuli-responsive macroscopic and microscopic 3D architectures. The revised introduction reads:

“Stimuli-responsive materials are key for active tunable systems.¹⁻⁴ In recent years, a large variety of material systems suitable for macroscopic⁵⁻⁷ and microscopic⁸⁻¹⁰ architectures have

been investigated and extensively reviewed.¹¹⁻¹⁵ Light as a local stimulus is of particular interest because light can readily be focused to small spots, allowing for controlled local responses. For applications in soft robotics, microfluidics, and biosciences,¹⁶⁻¹⁹ at least two conditions need to be fulfilled. First, the materials...

The following references are renumbered accordingly.

Reviewer #2

The authors have addressed quite a few of my comments and have improved the manuscript with additional experiments. I have the following questions/comments to add:

1. There are a few typos in the response letter and revised manuscript. E.g. p5, line 131. Please correct.

We have eliminated all typos that we could find.

2. The authors used a few different terms as to material design, such as single resist - two materials - bi-material - composite. Altogether, these terms are quite confusing for a non-specialist. The major difference is in their physical properties, but the same chemistry. Would it be possible to improve this for a better readability?

The fact that we use a single resist, which itself is a composite of several dissimilar ingredients, leading to different effective material properties after different levels of exposure by light, is a new and important aspect indeed. We feel, that this aspect has been described prominently at the end of our introduction. To further clarify, we have slightly modified this part to:

“...In this paper, we introduce a single photoresist. This photoresist is a composite based on *N*-isopropylacrylamide, the crosslinker *N,N*-methylenebisacrylamide, and a water-soluble photoinitiator. The novelty of this composite photoresist is twofold. First, the local properties in a three-dimensional (3D) microstructure can be tailored by the local exposure dose during 3D laser lithography (gray-tone lithography), opening the door to 3D hetero-microstructures from a single photoresist...”

3. Figure 3b: what are the possible reasons for the fluctuation? Is it due to inherent limitations with the material or the method to quantify curvature?

The fluctuations of the data points in Figure 3b were dominated by the analysis of the images. To improve our paper, we have changed three aspects:

1. We have repeated the analysis and have performed the fitting procedure several times. As a result, the fluctuations in the revised version of Figure 3b are smaller than they were previously. For convenience of the reviewer, this revised figure is reproduced below.
2. We have also added to this figure error bars, indicating the uncertainty resulting from the fitting procedure. It can be seen, that the fluctuations are within the error bars. Therefore, the figure shows no indication of material deterioration.

3. The meaning of the error bars is explained in the revised version of the main text:

“...To confirm the reversibility of the actuation process, we performed more than ten heating/cooling cycles on the sample and no changes in the optical images recorded after each step were detected. As an additional quantification, we manually fitted a circle to the optical micrographs for the respective temperatures. This procedure was repeated three times per image. The resulting mean value of the curvature, the inverse of the circle radius, and the corresponding standard deviation of the analysis (see error bars) are depicted in Figure 3b.

To demonstrate the large variety of possibilities with this approach we designed a micro-gripper that reversibly opens and closes on demand (see supplement Figure S2).

The simple bi-material beams and grippers exhibited a controllable and large-amplitude actuation. However, depending on the application, more complex actuation patterns are required. Thus, we realized hetero-structured beams which transform into different designed shapes as a response to the external stimulus...”

4. Limitation in simulation: It would be helpful for the authors to add an in-depth discussion with the limitations in the methodology selected for simulation.

We respond to points 4 and 5 together. See below.

5. Issue on Sample size: I do not think citing a Science article is sufficient to address this important point. The photoresist in the cited Metamaterial work was IP-S, which I think is totally different from pNIPAM in physical properties and mechanical response. Are there previous reports on the simulation of thermo-responsive gels like polyNIPAM? Is there non-linear elasticity with gel-like soft matter?

The science article was only referred to in our response letter, not in the manuscript itself. To improve our manuscript in this regard, we have expanded the corresponding discussion to:

“...These examples demonstrate that the response to the stimulus can be precisely controlled by local variation of the exposure dose in a hetero-structure composed of two largely different ingredients.

It is desirable to develop a simple theoretical model that is capable of predicting the actuation behavior of heterostructures. While advanced theoretical work on pNIPAM gels has been published,^{36,37} little is known on the mechanical properties of the materials emerging from the composite resist investigated here, fabricated using different exposure doses. Therefore, we employed the simplest possible continuum-mechanical description, namely linear Cauchy elasticity. Importantly, geometrical nonlinearities have been accounted for in the solver. In the spirit of reverse engineering, the input parameters were chosen such that the calculations obtained for the simplest bi-material beam (left structure in Figure 4b) closely resembled the experimental situation (left structure in Figure 4a). Details of the used numerical approach are given in the methods section. We found that the thermal expansion coefficient changes by one order of magnitude when increasing the exposure dose from 30 mW to 37.5 mW. For the same exposure dose variation, the Young's modulus changes by an order of magnitude, too (see methods section). Next, we applied the same equations and the same material parameters to model the three more complex structures shown in Figure 4. The comparison of all structures in Figure 4a and 4b reveals excellent agreement. Thus, the experimental findings are consistent with the simple theoretical model, hence there is no need for more advanced theoretical models. Our simple theoretical model even has predictive strength. However, our description is obviously not able to explain the molecular origin of the variation of the mechanical material properties as a function of the exposure dose. Such understanding is highly desirable to design and synthesize further new stimuli-responsive hydrogel systems, but is way beyond the scope of our paper.

In all results shown so far,...

Furthermore, we have revised the corresponding part of the methods section to:

“Numerical Analysis: The numerical analysis was performed by a finite-element approach using the commercial software COMSOL Multiphysics to solve the linear elastic Cauchy continuum-mechanics equations. Geometrical nonlinearities have been accounted for. For the bending calculations, the response of the structures to changes in temperature was introduced as a volumetric stress. The parameters used to model the two constituent materials A and B were chosen in such a way that the numerical results closely resemble the experimental situation in the simplest bi-material case...”

6. Considering the importance of photoinitiator for 2P-actuation, it would be important to include the content on this regard the main manuscript. Is the actuation efficiency dependent on the PI concentration? These control experiments are important.

This point is closely related to point 11 and we respond to both of them here. We have commented on this aspect in our previous response letter, but have not included this discussion in the manuscript. To improve our paper in this regard, we have moved the former Figure S4 to the new Figure 6 in the main paper. Furthermore, we have now included a discussion of these control experiments in the manuscript on page 15.

“... Thus, two-photon absorption allows to concentrate the excitation in all three spatial directions in 3D.

In addition, we have performed the following control experiment: We have fabricated PETA blocks containing a two-photon initiator (see methods section) next to the stimuli-responsive pNIPAM structures (see Figure 6). When the same femtosecond laser pulses as in the preceding paragraph are focused onto the PETA blocks, the pNIPAM structures again bend, whereas the PETA blocks exhibit no measurable response. This behavior is consistent with our interpretation that the bending of the pNIPAM structures is temperature-induced rather than being of structural origin. In yet additional control experiments, we have fabricated similar PETA blocks (not depicted), yet without a two-photon initiator (which is possible by directly exciting the HOMO-LUMO transition of the resist, albeit with reduced efficiency). We find negligible bending of the pNIPAM structures under otherwise similar conditions. We conclude, that the temperature increase is induced by two-photon absorption of the remaining photoinitiator (in this case Irgacure 819). Absorption spectra of the used photoinitiators can be found in literature.^{30,39,40}

However, only based on the experimental data it is challenging to extract reliable quantitative information about the temperature distribution on the substrate...”

7. Single-photon (or two-photon) absorption spectrum of preformed pNIPAM sample should be provided.

As argued in our response to the last point, the absorption spectra of the pNIPAM structures as well as the PETA structures are dominated by absorption via the corresponding photoinitiators. References to literature spectra have been given. These spectra are one-photon absorption spectra. Two-photon absorption spectra for these photoinitiators are not available in the literature.

8. Figure 2, please indicate why the PETA part is blue-colored.

To record 3D image stacks of the PETA structures via laser scanning fluorescence microscopy, the photoinitiator Irgacure 819 was replaced by the fluorescent photoinitiator 7-diethylamino-3-thenoylcoumarin (typically known as DETC). This initiator fluoresces in the green part of the visible spectrum. For esthetical reasons, we have colored these parts in turquoise. To improve our paper in this regard, we have modified the figure caption of Figure 2:

“...**Figure 2.** Stimuli-responsive pNIPAM valves in PETA microchannels. a) 3D reconstruction of experimental data recorded via confocal laser scanning microscopy. Two different color channels have been recorded, allowing to separate the fluorescence from the PETA with the green fluorescent DETC and that from pNIPAM with the red fluorescent rhodamine dye. The corresponding iso-intensity surfaces are colored in turquoise and gray, respectively. Upon heating the sample to 45 °C, the opening in the middle widens. This process is reversible when cooling the sample back down. b) Open area in the middle of the microchannel at 20 °C and 50 °C for multiple cycles of stimulation. We find no significant deterioration. c) Alternative design with an additional inner tube and two pNIPAM-tori. Complete closure of the microchannel can be achieved in a reversible manner...”

9. Page 16, line 366 in Methods section: Sentence „I819 was replaced by, is unclear. Is it a mixture of PI and dye? Or this dye was used as PI?

This question is related to the previous point. 7-diethylamino-3-thenoylcoumarin (DETC) is an efficient photoinitiator for two-photon polymerization which exhibits strong fluorescence. To clarify this aspect, we have added an explanation to the methods section:

“PETA-Resist: The resist was prepared by dissolving 20 mg Irgacure 819 in 980 mg of pentaerythritol triacrylate. Irgacure 819 was replaced by 7-diethylamino-3-thenoylcoumarin (DETC) for fluorescence imaging. DETC is an efficient photoinitiator for two-photon polymerization which also exhibits a strong fluorescence. This allowed us to record 3D image stacks of the fabricated PETA structures via LSM.”

10. Page 15, Methods section: Detailed experimental procedure for Figure 2 should be provided. It would be important for the authors to comment on the key fabrication steps.

To improve our manuscript in that regard, we added a new paragraph in the revised version of our methods section:

“**Fabrication of 3D Microstructures:** A commercial Direct Laser Writing setup (Photonic Professional GT, Nanoscribe GmbH) with a 25x, NA=0.8 oil immersion objective was used for fabrication. To increase the adhesion of the microstructures to the glass substrate we treated plasma-cleaned coverslips with 3-(trimethoxysilyl)propyl methacrylate (1 mM in toluene) for one hour and rinsed them afterwards in acetone and water.

Valve structures: The fabrication of the pNIPAM valves inside of PETA microchannels consisted of two consecutive steps. First, the PETA tubes, along with several alignment markers, were written on the cover glass and subsequently developed in a 1:1 mixture of isobutylmethylketone (MIBK) and isopropanol. Second, after drying, the pNIPAM-resist was drop cast onto the structures. The mentioned markers were crucial to ensure a precise lateral positioning of the valves inside the microchannels. The positioning along the third dimension was accomplished by using the built-in interface-finder. Finally, the structures were washed with acetone and developed in water.

Bi-material beam structures: Unless stated otherwise, the less (more) crosslinked beam was fabricated with a laser power at the back focal plane of 30 mW (37.5 mW) respectively and the post with 40 mW. The scanning speed was constant at 1 mm/s. After writing, the structures were rinsed with acetone and subsequently transferred into water for further development and storage. No post-curing treatment was applied.”

11. Supplement Figure S4 on control experiments should be included in the SI and discussed in the manuscript.

See our response to point 6 of reviewer #2.

Reviewer #3

The authors have addressed properly all comments except for two:

*1. The comment regarding the photo initiator reference-
The answer of the authors is incorrect, although the photo initiators are similar, Li-TPO stands for lithium 2,4,6-trimethylbenzoyldiphenyl phosphine oxide and not for lithium phenyl(2,4,6-trimethylbenzoyl)phosphinate that is used in the report.*

We thank the reviewer for pointing out this mistake. We have replaced the reference by the correct one:

Fairbanks, B.D., Schwartz, M.P., Bowman, C.N. & Anseth, K.S. Photoinitiated polymerization of PEG-diacrylate with lithium phenyl-2,4,6-trimethylbenzoylphosphinate: polymerization rate and cytocompatibility. *Biomaterials* **30**, 6702–6707 (2009)

*2. The comment regarding presenting Young's modulus values-
The authors did not provided results for this property. Why did they only estimate the values ?*

In the previous revised version of our manuscript we added a mechanical analysis via atomic force microscopy. With this method it is possible to derive the Young's modulus for a specific set of fabrication parameters. However, when we write bi-material structures with one beam at high laser power and then an adjacent one with lower laser power, the former is influenced by the latter. For this reason, it is not possible to simply analyze the two materials one by one.

REVIEWERS' COMMENTS:

Reviewer #2 (Remarks to the Author):

The authors revised the manuscript with in-depth discussion. But I am still confused by the appropriateness of term 'composite'. A gel-formulation (gel precursors, initiator) without varied laser curing is not a composite material (ref. 10.1126/science.aav7390).

Suggestion:

„ In this paper, we introduce a single photoresist based on poly (N-isopropylacrylamide) to create a composite material with spatially-resolved mechanical properties. „

I recommend it for publication in Nature Commu after this point is addressed.

Reviewer #3 (Remarks to the Author):

The authors addressed properly the comments.

Response to the comments of the reviewers

In what follows, *we repeat the comments of the reviewers in red and italic*, respond to them in green, repeat passages from the original manuscript in black, and highlight changes made in the revised version of our manuscript in blue.

Reviewer #2:

The authors revised the manuscript with in-depth discussion. But I am still confused by the appropriateness of term 'composite'. A gel-formulation (gel precursors, initiator) without varied laser curing is not a composite material (ref. 10.1126/science.aav7390).

Suggestion:

„ In this paper, we introduce a single photoresist based on poly (N-isopropylacrylamide) to create a composite material with spatially-resolved mechanical properties. „

I recommend it for publication in Nature Commu after this point is addressed.

We thank the reviewer for this positive assessment of our work and the recommendation for publication.

To improve our manuscript we have replaced the misleading term “composite” and changed the respective sentences to:

“...In this paper, we introduce a single photoresist based on *N*-isopropylacrylamide, the crosslinker *N,N'*-methylenebisacrylamide, and a water-soluble photoinitiator. The advantages of this photoresist are twofold. First, the local properties in a three-dimensional...”

and

...While advanced theoretical work on pNIPAM gels has been published,^{36,37} little is known on the mechanical properties of the materials emerging from the photoresist system investigated here, fabricated using different exposure doses. Therefore, we employed the simplest possible...

Reviewer #3:

The authors addressed properly the comments.

We thank the reviewer for his helpful and constructive comments to improve our manuscript.